# Causal-PIK:
# Causality-based Physical Reasoning with a Physics-Informed Kernel

**Carlota Parés-Morlans** [1]  **Michelle Yi** [1]  **Claire Chen** [1]  **Sarah A. Wu** [2]  **Rika Antonova** [1 3]  **Tobias Gerstenberg** [2]
**Jeannette Bohg** [1]

## Abstract

Tasks that involve complex interactions between objects with unknown dynamics make planning before execution difficult. These tasks require agents to iteratively improve their actions after actively exploring causes and effects in the environment. For these type of tasks, we propose Causal-PIK, a method that leverages Bayesian optimization to reason about causal interactions via a Physics-Informed Kernel to help guide efficient search for the best next action. Experimental results on Virtual Tools and PHYRE physical reasoning benchmarks show that Causal-PIK outperforms state-of-the-art results, requiring fewer actions to reach the goal. We also compare Causal-PIK to human studies, including results from a new user study we conducted on the PHYRE benchmark. We find that Causal-PIK remains competitive on tasks that are very challenging, even for human problem-solvers.

## 1. Introduction

Consider trying to solve a physical reasoning puzzle from the Virtual Tools benchmark, like those shown in Figure 1. Your goal is to have the red ball end up in the green region. To make that happen, you need to choose one of the three objects from the top left, place it somewhere in the scene, and let gravity and causality do the rest. Which object would you choose and where would you place it? Research in cognitive science suggests that humans solve tasks like these by building internal models of the physical world (Battaglia et al., 2013; Smith et al., 2018; Ota et al., 2021; Zhou et al.,

2023). These models encode our causal understanding of the domain – our beliefs about how one event leads to another.

While predicting the exact sequence of events is difficult, our physical intuition allows us to estimate the immediate causal effects of our actions, such as which object will move and in which direction. In addition, we rapidly learn from the outcomes of previous actions (Allen et al., 2020). If the red ball almost landed in the goal region, we would slightly change our action and try again. If it was way off, we might try something completely different. Inspired by people's rapid learning in such tasks, we develop a method that leverages physical intuitions to efficiently solve physical reasoning tasks.

Researchers have developed several benchmarks for evaluating the physical reasoning capabilities of AI agents (Melnik et al., 2023). These benchmarks range from predicting the stability of a stack of blocks from an image to recognizing violations of physical principles. In this work, we focus on single-intervention physical reasoning tasks, where an agent chooses what action to take at the beginning of an episode and then observes the action's effects on the environment. We focus on the Virtual Tools (Allen et al., 2020) and PHYRE (Bakhtin et al., 2019) benchmarks, which despite looking deceivingly simple in their 2D form, are very hard to solve even for humans (Allen et al., 2020). These tasks involve complex physical interactions between objects, which without complete information about the environment dynamics make it impossible for agents to plan the exact solution without active exploration. Agents must interact with the environment to actively obtain information about the object dynamics. The main challenge is to reason about the causal consequences of one's actions, to leverage this knowledge in order to make effective decisions about what actions to try, and to rapidly learn from previous attempts.

We propose Causal-PIK, a method that leverages physical intuition and causality to solve single-intervention physical reasoning tasks in as few trials as possible. Our method uses Bayesian optimization to reason about causality via a Physics-Informed Kernel to obtain an expressive posterior distribution over the environment dynamics. This allows an agent to efficiently explore the search space and intelligently

---

[1]Department of Computer Science, Stanford University, CA, USA [2]Department of Psychology, Stanford University, CA, USA [3]Department of Computer Science and Technology, University of Cambridge, Cambridge, UK. Correspondence to: Carlota Parés-Morlans <cpares@stanford.edu>.

*Proceedings of the 42$^{nd}$ International Conference on Machine Learning*, Vancouver, Canada. PMLR 267, 2025. Copyright 2025 by the author(s).

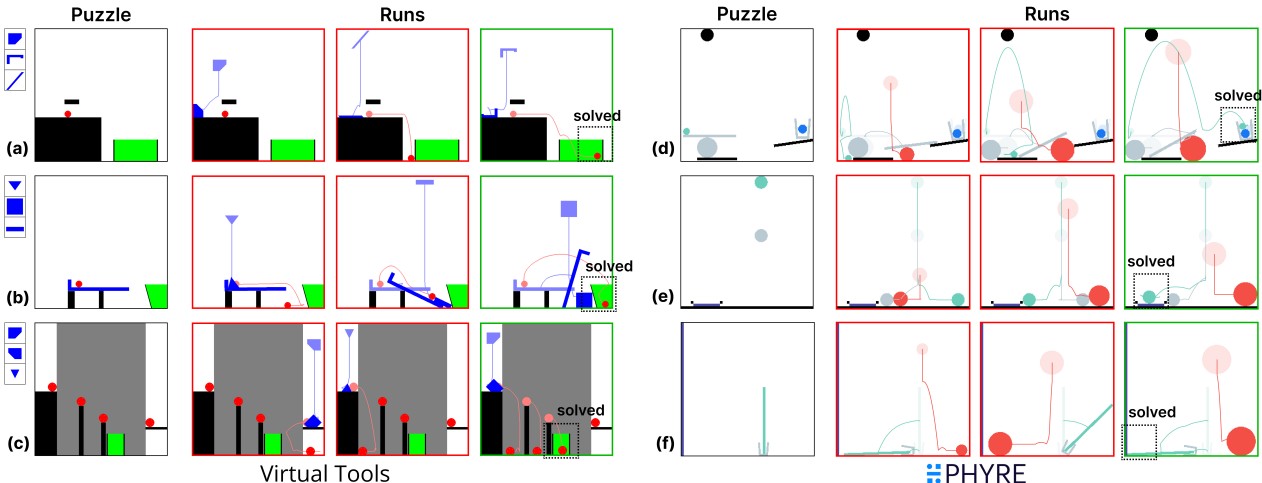

*Figure 1.* Example puzzles from the Virtual Tools (Allen et al., 2020) and PHYRE (Bakhtin et al., 2019) benchmarks. For Virtual Tools, the objective is to place one of the blue action objects from the left to have a red ball fall into the green area. For PHYRE, the objective is to place a red ball of variable radius in the environment to have the green and blue objects touch for at least 3 seconds. The runs highlighted in red are failed attempts and the runs highlighted in green are successful attempts.

select what actions to evaluate next. Causal-PIK minimizes the number of physical interactions required. We demonstrate that Causal-PIK significantly outperforms state-of-the-art models on the single-intervention physical reasoning tasks from the Virtual Tools (Allen et al., 2020) and PHYRE (Bakhtin et al., 2019) benchmarks, finding solutions in the fewest number of attempts.

## 2. Related Work

**Physical Reasoning:** Researchers have developed several benchmarks towards the goal of improving physical reasoning capabilities in machine-learning models (Melnik et al., 2023). We evaluate Causal-PIK on the PHYRE (Bakhtin et al., 2019) and Virtual Tools (Allen et al., 2020) benchmarks. Both benchmarks use a physics simulator and capture a variety of complex interaction mechanisms. Furthermore, both contain a large set of puzzle variations, allowing us to study generalization across diverse environments.

Both PHYRE and Virtual Tools have inspired several methods for solving physical reasoning puzzles. A number of works (Harter et al., 2020; Girdhar et al., 2020; Ahmed et al., 2021; Qi et al., 2021; Li et al., 2022) that evaluate on PHYRE leverage forward prediction models, also called dynamics models or world models, to predict the outcomes of actions. They use dynamics models to score actions and then execute the most promising actions until one succeeds. Like these works, Causal-PIK also uses a dynamics model to reason about the action outcomes. However, unlike these prior methods, Causal-PIK uses observations from previous trials to inform future action selection via Bayesian optimization. Instead of using a dynamics model to directly choose actions, we use dynamics predictions to instill

physical intuition into kernel updates during Bayesian optimization. We show that our method solves PHYRE in fewer attempts than any of these prior methods.

The 'Sample, Simulate, Update' model (SSUP) (Allen et al., 2020) attempts to solve the Virtual Tools benchmark. It samples actions from an object-based prior, simulates the sampled actions in a noisy physics engine to find the best action to try, executes the action, and updates the model's belief using information from both simulation and execution. SSUP uses a Gaussian mixture model for guidance, but it doesn't embed information about how actions are related via their effects, making SSUP search less efficient. In contrast, our work builds on active learning methods, such as Bayesian optimization, and successfully incorporates additional guidance for action selection by introducing a Physics-Informed Kernel. This significantly lowers how many trials are needed to succeed.

**Bayesian optimization:** Causal-PIK uses Bayesian optimization (BO), a global search method for optimizing black-box functions (Shahriari et al., 2015; Wang et al., 2023). Similar to how we use BO to efficiently find puzzle-solving actions, BO has been used in robotics to reduce the number of trials needed to be run on real robots (Feng et al., 2015; Calandra, 2017; Jaquier et al., 2020; Berkenkamp et al., 2023). Some prior works proposed to further increase data-efficiency by incorporating simulation-based information (Antonova et al., 2017; Marco et al., 2017; Antonova et al., 2019). However, they mostly considered continuous parametric controllers, while we need to accommodate a hybrid continuous/discrete action space.

**Inferring Causality:** For the type of tasks we study in this work, the ability to reason about the outcome of an interac-

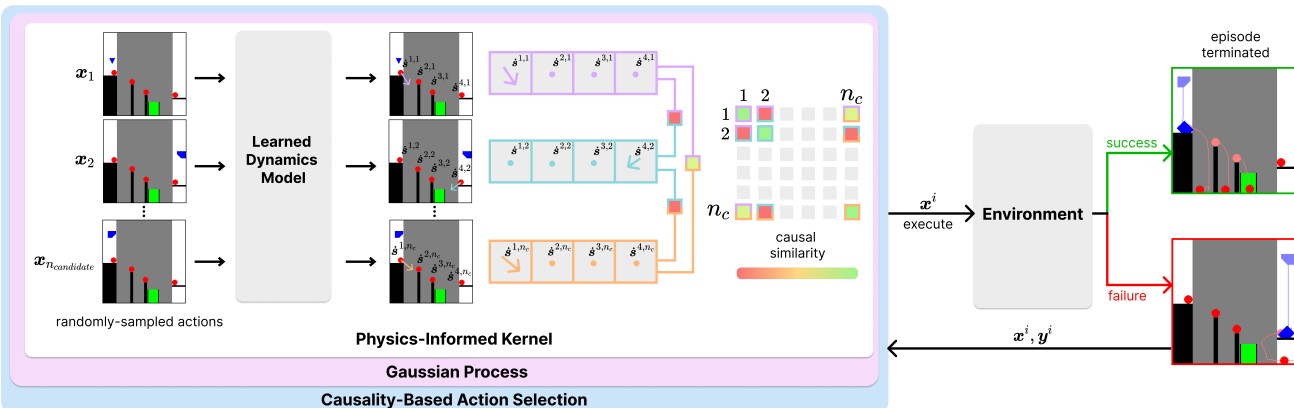

*Figure 2.* An overview of Causal-PIK. The Causality-Based Action Selection module proposes the best action to execute based on the predictions from the Gaussian Process. These predictions are generated considering the rewards from previous actions and the causal similarities between candidate actions computed by the Physics-Informed Kernel. These similarities are derived from the individual effects of actions, as predicted by the learned dynamics model. They are expressed as the state change of object $O$ caused by action $a$, denoted as $\dot{\boldsymbol{s}}^{O,a}$.

tion is critical (Ullman et al., 2018; Bramley et al., 2018). Some works aim to reason about causality by focusing on the Violation of Expectations (VoE) paradigm (Smith et al., 2019; Riochet et al., 2021), where models learn physical plausibility by observing scenes that either follow or violate intuitive physics rules. These models can then judge if new scenes are plausible or not based on past observations. Other works utilize deep neural networks to predict the outcome of interactions from frames (Duan et al., 2022; Ye et al., 2018). If an agent can reason about the outcome of inter-actions before actions are taken, then the reasoning can be used to guide smarter action selection, which is beneficial when taking actions in the real world is costly. (Battaglia et al., 2013) and (Wu et al., 2017) introduce methods that conduct physical reasoning with the help of simulation after constructing a representation of the physical world from visual inputs.

## 3. Method

**Problem Formulation:** We consider single-intervention physical reasoning tasks where an agent executes an action at the beginning of an episode, observes the action's effects on the environment over $T$ episode steps, and then receives a score at the final timestep. After each episode, the environment resets, and the agent may attempt the task again by trying another action. Formally, on each attempt, an agent executes an action $\boldsymbol{x}$ at the first state $\boldsymbol{s}_0$ and observes this action's effect on the environment's state at the remaining timesteps, $\boldsymbol{s}_t$ for $t = 1, 2, ..., T$. The state evolves according to the environment's unknown dynamics, which we denote as the function $\mathbb{D}(\boldsymbol{s}_0, \boldsymbol{x})$, to give $\boldsymbol{s}_{1,...T} = \mathbb{D}(\boldsymbol{s}_0, \boldsymbol{x})$. At the end of the episode, the agent receives a score $y$, which is computed based on the observed states $\boldsymbol{s}_{1,...T}$ using a score function $\mathbb{S}(\boldsymbol{s}_{1,...T})$: $y = \mathbb{S}(\boldsymbol{s}_{1,...T})$. For a given initial

state $\boldsymbol{s}_0$, the future states $\boldsymbol{s}_{1,...T}$ are a function of the action, so we can re-write the score function as a function of $\boldsymbol{x}$: $y = \mathbb{S}(\mathbb{D}(\boldsymbol{s}_0, \boldsymbol{x})) = f(\boldsymbol{x})$.

Given an initial state $\boldsymbol{s}_0$, we aim to solve the task by finding the action that achieves the highest score, namely maximizing $f(\boldsymbol{x})$, in as few attempts as possible. However, the task's unknown dynamics make it difficult to find optima of $f(\boldsymbol{x})$. Therefore, we use Bayesian optimization to find actions that maximize the score.

**Bayesian Optimization (BO):** The goal of BO is to find the optimal $\boldsymbol{x}^*$ that maximizes a given objective $f(\boldsymbol{x})$. The objective captures characteristics of the desired outcome, e.g., in our setting, the task score. BO begins with a prior that expresses uncertainty over $f(\boldsymbol{x})$. After evaluating an $\boldsymbol{x}$, BO constructs a posterior based on the observed data $y$ obtained so far. It then uses an auxiliary acquisition function to pick the next $\boldsymbol{x}$ to evaluate, taking into account both the posterior mean and covariance. A Gaussian process (GP) is commonly used to model the objective function: $f(\boldsymbol{x}) \sim GP(\mu(\boldsymbol{x}), k(\cdot, \cdot))$. Its kernel function describes the covariance between the objective values of any pair of input points: $k(\boldsymbol{x}_1, \boldsymbol{x}_2) = cov(f(\boldsymbol{x}_1), f(\boldsymbol{x}_2))$.

A common choice for the kernel is the Radial Basis Function (RBF): $k_{RBF}(\boldsymbol{x}_1, \boldsymbol{x}_2) = \sigma_k^2 \exp(-\frac{1}{2}(\boldsymbol{x}_1 - \boldsymbol{x}_2)^T \text{diag}(\boldsymbol{l})^{-2}(\boldsymbol{x}_1 - \boldsymbol{x}_2))$, where $\sigma_k^2$, $\boldsymbol{l}$ can be tuned automatically. The kernel plays a crucial role in BO, as it encodes an inductive bias about the properties of the underlying objective function to choose the next $\boldsymbol{x}$. The RBF kernel, in particular, assumes that values geometrically closer in the input space have a higher correlation (are more similar) than values geometrically farther in the input space.

Selecting an appropriate kernel significantly impacts the accuracy of the objective function approximated by the GP,

thus influencing the performance of BO. We develop a kernel that encodes an intuition of causality and physics, which we show solves physical reasoning tasks in fewer attempts than the RBF kernel. In the following section, we describe our full physics-informed BO algorithm Causal-PIK.

### 3.1. Causal-PIK

Causal-PIK uses BO and reasons about causality with a Physics-Informed Kernel to solve physical reasoning tasks. It iteratively proposes actions and uses their observed outcomes, in conjunction with an intuition of physics, to inform future action proposals. We encode physics intuition, such as effects of geometry, mass, and momentum, into the BO framework via the learned Physics-Informed Kernel used to build the GP that approximates the objective function. We will show that incorporating knowledge of physics through the kernel function allows BO to more quickly discover promising regions of the action space.

---

**Algorithm 1** Causal-PIK

1: Attempt $i = 0$
2: $X = \{\boldsymbol{x}_1^{\text{init}}, ..., \boldsymbol{x}_{\boldsymbol{n}_{\text{initial}}}^{\text{init}}\}$, $y = \{\hat{y}_1^{\text{init}}, ..., \hat{y}_{n_{\text{initial}}}^{\text{init}}\}$
3: **while** success == False **do**
4:     $gp \leftarrow$ **GP**$(X, y, \texttt{PhysicsInformedKernel})$
5:     $\boldsymbol{x}^i \leftarrow \texttt{CausalityBasedActionSelection}(gp)$
6:     $y^i, \text{success} \leftarrow \texttt{Execute}(\boldsymbol{x}^i)$
7:     $X \leftarrow \text{concat}(X, \boldsymbol{x}^i)$, $y \leftarrow \text{concat}(y, y^i)$
8:     $i \mathrel{+}= 1$
9: **end while**

---

Algorithm 1 outlines Causal-PIK. First, we construct initial sets of actions $X$ and scores $y$ to initialize the GP prior using a probabilistic intuitive physics engine (Battaglia et al., 2013). After initializing $X$ and $y$, we begin the BO procedure, iteratively updating the physics-informed GP surrogate function, choosing the next action and executing it until the puzzle is solved.

**GP Initialization (Alg. 1 L2)**: To initialize the GP for both Virtual Tools and PHYRE, we use $n_{initial} = 9$ initial data points. This is the same number of initial data points that our Virtual Tools baseline SSUP (Allen et al., 2020) uses to initialize their method. SSUP found that this value provided a good trade-off between the number of initial points, total attempts required, and convergence time. Since we use the same initialization framework as SSUP, we expect their parameter analysis to extend to our results. Importantly, like SSUP, we treat these initial noisy rollouts as warm-up samples that do not count towards the total attempt count.

We choose the initial points following the same approach as SSUP: randomly select a dynamic object from the environment and sample a point from a Gaussian distribution centered at the object's center. As a result, each puzzle attempt has a unique set of $n_{initial}$ points. This heuristic

helps the GP build a noisy prior that includes different areas.

**Physics-Informed GP Update (Alg. 1 L4)**: On the $i$th attempt, we update the GP surrogate function with all attempted actions $X$ and their observed outcomes $y$ using our learned Physics-Informed Kernel. Conceptually, the Physics-Informed Kernel encodes similarities and differences between actions based on the predicted effects they will have on the environment. Using this kernel to construct the GP helps the GP use the observed outcomes of already-attempted actions to more accurately predict the outcomes of unexplored actions. We derive the Physics-Informed Kernel in Section 3.2.

**Causality-Based Action Selection (Alg. 1 L5)**: On the $i$th attempt, we select the next most promising action $\boldsymbol{x}^i$ by finding the action that maximizes an Upper Confidence Bound (UCB) acquisition function. This acquisition function considers the mean and uncertainty of the physics-informed GP. First, we use a Sobol sequence generator to sample a set of $n_{\text{candidate}} = 500$ candidate actions. Then, we evaluate the acquisition function at each of these $n_{\text{candidate}}$ actions. Adopting the intuitive physics procedure proposed by Allen et al., we approximate the outcome of the $n_{\text{best}} = 5$ candidate actions with the highest acquisition function values using a probabilistic simulation of the task. As per (Allen et al., 2020), approximating the outcomes of actions mimics how humans use mental representations to imagine the potential effects of actions before committing to an action. Finally, from this set of $n_{\text{best}}$ actions, we select the action with the highest expected outcome as the next action.

**Action Execution (Alg. 1 L6)**: We execute the selected action $\boldsymbol{x}^i$, observe its outcome, and compute the corresponding reward $y^i$. If $\boldsymbol{x}^i$ solves the task, the algorithm terminates. If $\boldsymbol{x}^i$ fails to solve the task, we append it along with the score $y^i$ to $X$ and $y$, respectively.

### 3.2. Physics-Informed Kernel

In this section, we describe the Physics-Informed Kernel used to construct the GP in BO (Alg. 1 L4). We construct the Physics-Informed Kernel such that it captures two types of physics intuition that are important for solving physical reasoning tasks. The first type is the ability to reason about the causal effect of individual actions, or in other words, the changes that single actions induce in the environment. To capture this, we train a dynamics model to predict future states of the environment given an action and initial state. The second type of physics intuition that we encode into the kernel is the ability to reason about causal similarity, namely which actions cause similar changes in the environment. To capture this, we define a function that computes how similar two actions are based on their causal effects.

As shown in Fig. 2, the Physics-Informed Kernel function

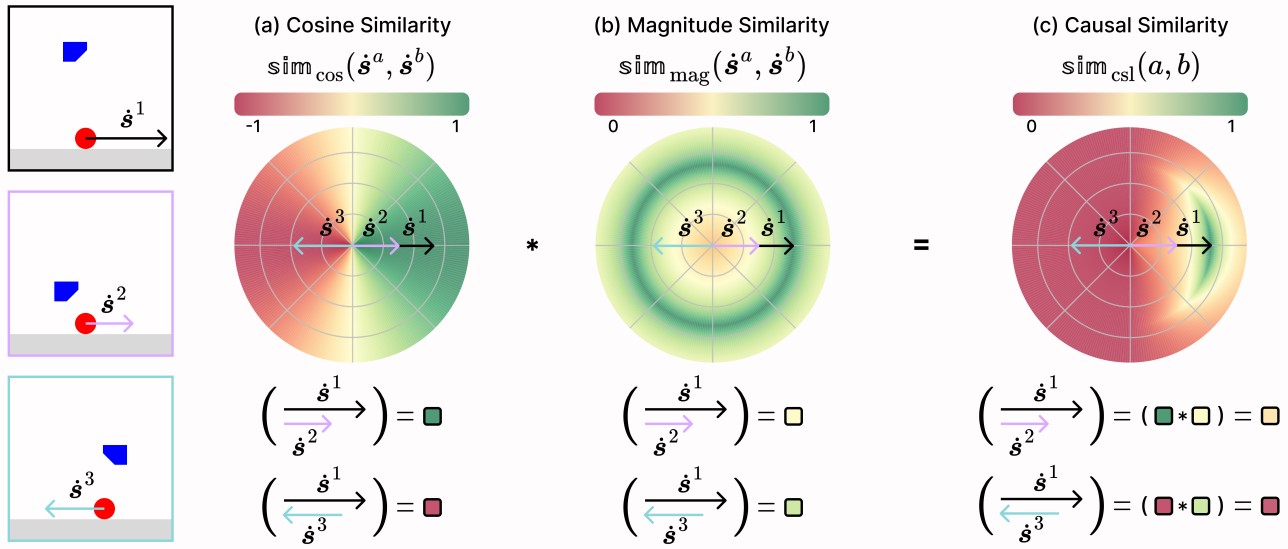

*Figure 3.* Illustration of how causal similarity is computed. We obviate the object superscript as there is only one object (red ball) in this example. (a): Cosine similarity (Eq. 3). $\dot{s}^1$ has a high cosine similarity with $\dot{s}^2$ as they point in the same direction. $\dot{s}^3$ points in the opposite direction of $\dot{s}^1$, obtaining a negative similarity value. (b): Magnitude similarity (Eq. 4). $\dot{s}^2$ has a magnitude that is 40% smaller than $\dot{s}^1$, receiving a low magnitude similarity score. $\dot{s}^3$ has a magnitude that is very close to $\dot{s}^1$, obtaining a high similarity value. (c) Final causal similarity combining cosine and magnitude similarities (Eq. 6). We obtain a medium similarity score for $\dot{s}^2$ relative to $\dot{s}^1$ and low score for $\dot{s}^3$ relative to $\dot{s}^1$.

first uses the learned dynamics model to predict action outcomes. It then uses these dynamics model predictions to predict the similarity between actions. Using the predictions from the learned dynamics model to predict action similarity allows the Physics-Informed Kernel function to model the correlation between actions. In the following sections, we describe the learned dynamics model and the causal similarity function. Then, we show that the causal similarity function is a valid kernel function.

**Predicting Causal Effects via Learned Dynamics**

First, we train a dynamics model $\hat{\mathbb{D}}$ to predict the causal effects of individual actions. The physical reasoning tasks we consider involve placing an action object into an environment with $D$ dynamic objects. The action object and all dynamic objects move in the environment according to the unknown world's dynamics. We train the model to predict the effects that individual actions $x$ have on these dynamic objects. Formally, at each timestep $t$ of an episode, the environment's state vector $s_t$ consists of the state of the action object, $s_t^A$, and the states of every dynamic object, $s_t^1, s_t^2, ..., s_t^D$. For a given action $x$ and initial environment state $s_0$, the dynamics model $\hat{\mathbb{D}}$ predicts the state at the next $n_{\text{pred}}$ timesteps:

$$[s_1 \quad ... \quad s_{n_{\text{pred}}}] = \hat{\mathbb{D}}(s_0, x). \quad (1)$$

**The Causal Similarity Function**

We use predictions from the learned dynamics model $[s_1 \quad ... \quad s_{n_{\text{pred}}}]$ to compute the causal similarity of actions,

which captures how similar actions are based on their ability to cause similar outcomes in the environment. To achieve this, we begin by finding the timestep of the first causal event, where the action object interacts with any of the $D$ dynamic objects, which we call $t_{\text{event}}$. If the action object does not interact with any dynamic object during the $n_{\text{pred}}$ steps, we set $t_{\text{event}}$ to be the initial timestep, 0. Next, we quantify the causal effect of an action on a dynamic object $O$ based on its predicted motion as

$$\dot{s}^O = \frac{s_{(t_{\text{event}}+\Delta t)}^O - s_{t_{\text{event}}}^O}{\Delta t}, \quad (2)$$

which computes the state change of object $O$ between $t_{\text{event}}$ and $\Delta t$ timesteps after $t_{\text{event}}$.

To quantify similarity of two actions $a$ and $b$, we first look at the difference in the state changes, predicted by the dynamics model, they cause to a given object $O$. In computing this per-object similarity, we account for both the difference in the directions of the state changes caused by each action on the object (the cosine similarity),

$$\text{sim}_{\cos}(\dot{s}^{O,a}, \dot{s}^{O,b}) = \frac{\dot{s}^{O,a} \cdot \dot{s}^{O,b}}{||\dot{s}^{O,a}|| \, ||\dot{s}^{O,b}||} \in [-1, 1], \quad (3)$$

and the difference in the magnitudes of the state changes caused by each action on the object,

$$\text{sim}_{\text{mag}}(\dot{s}^{O,a}, \dot{s}^{O,b}) = \frac{1}{1 + |\,||\dot{s}^{O,a}|| - ||\dot{s}^{O,b}||\,|} \in [0, 1], \quad (4)$$

where $\dot{s}^{O,a}$ and $\dot{s}^{O,b}$ are the state changes of object $O$ caused by actions $a$ and $b$, respectively (as computed by Eq. 2). Combining these, the full per-object similarity of actions $a$ and $b$ for an object $O$ is

$$\text{sim}_{\text{obj}}(O, a, b) = \max \big[ \\ 0, \quad \text{sim}_{\cos}\big(\dot{s}^{O,a}, \dot{s}^{O,b}\big) \, \text{sim}_{\text{mag}}\big(\dot{s}^{O,a}, \dot{s}^{O,b}\big) \\ \big] \in [0, 1]. \tag{5}$$

Finally, we define the full causal similarity metric between actions $a$ and $b$ as:

$$\text{sim}_{\text{csl}}(a, b) = \frac{1}{D} \sum_{O=1}^{D} [\text{sim}_{\text{obj}}(O, a, b)] \\ \cdot exp\left( \left[ \frac{1}{D} \sum_{O=1}^{D} \text{sim}_{\text{obj}}(O, a, b) \right] - 1 \right). \tag{6}$$

The above computes the mean per-object similarity over all $D$ dynamic objects scaled by the exponentiated mean to accentuate differences between state changes caused by actions. Figure 3 provides a visual example of Equations (3)-(6) and shows how the causal similarity metric captures both directional and magnitude similarity.

**From Causal Similarity to Physics-Informed Kernel**

The causal similarity function in Equation 6 uses predicted state changes to quantify the correlation between actions that have similar causal effects. Consequently, these scores possess significant potential to be used as the kernel function of a GP that models the objective function of physical reasoning tasks. For a function to be a valid kernel, it must satisfy two fundamental properties: symmetry and positive semi-definiteness. Analyzing Equations (3)-(6), the reader can verify that $\text{sim}_{\text{csl}}(a, b) = \text{sim}_{\text{csl}}(b, a)$, thereby fulfilling the symmetry criterion. Furthermore, for any arbitrary pair of actions $a$ and $b$, their similarity score $\text{sim}_{\text{csl}}(a, b)$ is always non-negative. As such, causal similarity satisfies both criteria of valid kernel function, so we define the Physics-Informed Kernel used in Algorithm 1 Line 4 as $\text{sim}_{\text{csl}}(a, b)$.

### 3.3. Causal-PIK objective function

We use Causal-PIK to find an optimal action $x$ that maximizes the objective function $f(x)$. On each attempt, we use an objective function $f(x)$ that quantifies the progress that executing action $x$ makes towards a goal state $s^g$. We find that for tasks with complex dynamics, it is important for the objective function to capture how close an action gets to reaching the goal state at *any* timestep in the $T$-timestep long episode, not just at the final timestep. This insight aligns with the concept of "almost" reaching a goal, introduced by Gerstenberg et al. (Gerstenberg & Tenenbaum, 2016). To capture this, we use the closest distance

$d_c = \min_{t=1,\dots,T} \text{dist}(s_t, s^g)$ to the goal state achieved at *any* timestep $t$ in an episode to compute the objective function $f(x)$ (where $\text{dist}(\cdot)$ denotes a distance function):

$$f(x) = \begin{cases} \left(1 - \dfrac{d_c}{\text{dist}(s_0, s^g)}\right) \exp(\beta d_c) & \text{if } d_c < \text{dist}(s_0, s^g) \\ 0 & \text{otherwise} \end{cases} \tag{7}$$

### 3.4. Leveraging Causality for Efficient Sample Learning

Unlike methods that learn only from direct observations, our approach leverages physical reasoning to infer the outcomes of untested actions that are predicted to share the same causal effect as observed ones. As a result, a single rollout allows our model to update its belief not just about the executed action, but also about actions that are predicted to produce a similar physical outcome. This significantly enhances sample efficiency and enables reasoning about alternative scenarios without exhaustive exploration.

Our Physics-Informed Kernel explicitly encodes causal dependencies between actions and their physical consequences. In contrast to standard approaches that tend to cluster actions based on proximity in the feature space, Physics-Informed Kernel compares them by assessing their impact on the environment. Using counterfactual reasoning, the kernel distinguishes between causal effects – those directly attributable to actions – and confounding factors arising from the dynamics of the environment.

This distinction is achieved by evaluating action outcomes against a counterfactual baseline where no action object is placed in the environment. In Equation 7, the shortest observed distance $d_c$ is normalized by the counterfactual baseline distance $\text{dist}(s_0, s^g)$. Similarly, when computing the causal similarity of actions in Equation 6, the counterfactual baseline helps cluster actions that have no effect on the environment and emphasizes the attributable effects of other actions.

## 4. Experimental Setup

**Benchmarks:**

Virtual Tools: We test Causal-PIK on the 20 original puzzles from the Virtual Tools benchmark. These puzzles involve placing an action object in a 2D dynamic physical environment to guide a red ball into the green goal area. Once an action object is placed, gravity is activated, and its effect can be observed.. The three-dimensional action space consists of: $x_{pos} \in [0, 600]$, $y_{pos} \in [0, 600]$, and $action_{obj} \in \{tool_1, tool_2, tool_3\}$, resulting in a total of 1,080,000 possible actions.

PHYRE: We test Causal-PIK on the 25 puzzles from the BALL tier in PHYRE, following the cross-generalization

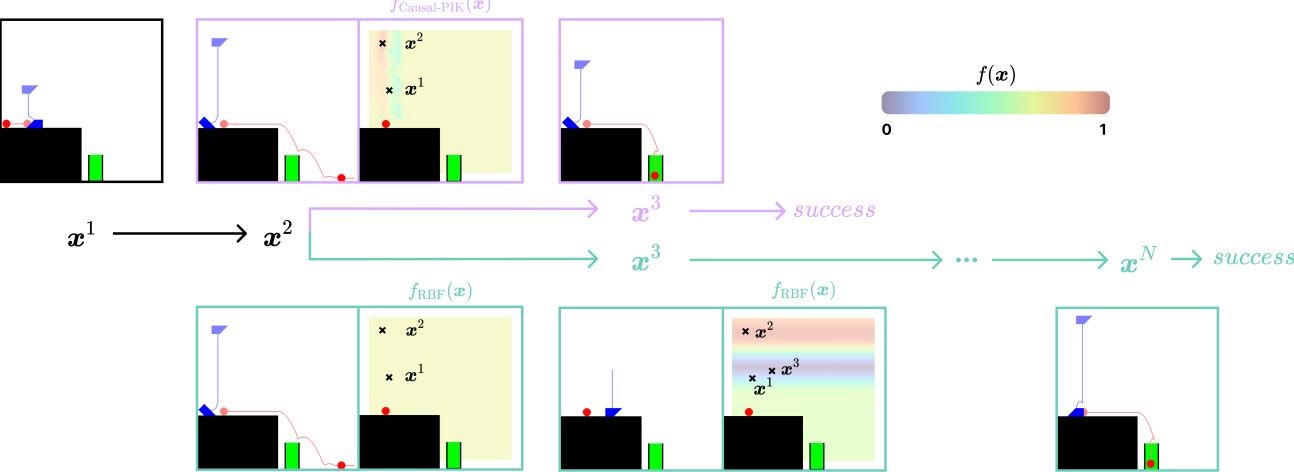

*Figure 4.* Comparison of posterior distributions $f(\boldsymbol{x})$ generated by Causal-PIK and a BO agent with an RBF kernel. Given the same initial observations $\{\boldsymbol{x}^1, \boldsymbol{x}^2\}$, Causal-PIK produces a more expressive posterior, effectively clustering actions based on their causal effects - the red area contains those actions predicted to move the ball closer to the goal while the blue area contains those predicted to push it further away. In contrast, the BO agent with the RBF kernel requires more iterations to develop an informative posterior.

setting. These puzzles require selecting the radius of a ball and placing it in a 2D dynamic physical environment to have the blue and green objects stay in contact for at least 3 seconds. As in Virtual Tools, gravity is activated upon placement, after which the action's effect is observed. The three dimensional action space consists of: $x_{pos} \in [0, 256]$, $y_{pos} \in [0, 256]$, and $action_{obj} \in [2, 32]$, yielding a total of 2,555,904 possible actions.

**Baselines and Ablations:** For Virtual Tools, we compare Causal-PIK against SSUP (Allen et al., 2020), the state-of-the-art method for this benchmark. We also compare against the two naive baselines from Allen et al., RAND and DQN. For PHYRE, we compare Causal-PIK against three categories of prior work. The first category consists of a method that uses a reduced action space of size 1,000 (Girdhar et al., 2020). The second category consists of methods that use a reduced action space of size 10,000 (Ahmed et al., 2021; Qi et al., 2021). Critically, by reducing the action space from ~2.5 million actions, these methods simplify the problem dramatically. The third category consists of methods that operate over the full action space (Harter et al., 2020), as we do. Additionally, we report results for three naive baselines from the original PHYRE paper (Bakhtin et al., 2019): RAND, MEM, and DQN. Finally, to assess the impact of the Physics-Informed Kernel, we conduct an ablation study by replacing it with a standard RBF kernel. This baseline highlights the expressiveness of the Physics-Informed Kernel, particularly in tasks with complex dynamics, where capturing underlying physical principles is key to finding solutions efficiently.

**Human Baseline:** In addition to model baselines, we compare Causal-PIK against a human baseline for both benchmarks. A human study with Virtual Tools was conducted by Allen et al. (2020). Here, we conducted a human experiment with PHYRE by recruiting $n = 50$ participants from Prolific and asking each participant to solve one variation of each of the 25 puzzles. Participants had 10 attempts to solve each puzzle by using the mouse to draw and place a valid action object. As with the model setup, each puzzle was initially presented as a freeze-frame, and then gravity was activated once the participant submitted their action (see Appendix B for details). The experiment was approved by the Stanford Institutional Review Board (IRB).

**Evaluation Metric:** We measure the performance of our method using the AUCCESS metric introduced by Bakhtin et al.. This metric aggregates the success percentages via a weighted average, placing more emphasis on solving tasks with fewer attempts. AUCCESS is computed as follows:

$$AUCCESS = \frac{\sum_k w_k \cdot s_k}{\sum_k w_k}, \text{ where}$$
$$w_k = log(k+1) - log(k), \text{ for } k \in \{1, ..., \text{MAX\_ATT}\}$$
$$s_k : \text{percentage of tasks solved within } k \text{ attempts} \quad (8)$$

We compare overall AUCCESS scores for each model, as well as correlation between scores for humans and models across individual puzzles.

**Dynamics Model:** We use Region Proposal Interaction Networks (Qi et al., 2021) as the architecture. We train the model on puzzles that share underlying physical concepts similar to the test puzzles. For Virtual Tools, we train the model on 10 variations for each original puzzle. For PHYRE, for each of the 10-fold splits from Bakhtin et al., we train a model exclusively on the fold's training set, ensuring that Causal-PIK is tested on previously unseen puzzles. At inference time, an image of the test puzzle with the action

embedded and the initial bounding boxes of all dynamic objects in the scene are fed into the model. The model outputs the bounding boxes of the next $n_{pred}$ time steps. We set $n_{pred}$ to capture the initial steps, but not the full rollout. The choice of $n_{pred}$ helps the dynamic model focus on learning the immediate causal effect of an action, which is then used to compare the similarity between actions. For further details, refer to Appendix A.

## 5. Results and Discussion

**Virtual Tools:** We evaluate Causal-PIK on the 20 Virtual Tools puzzles with 100 tests per puzzle. For each test, we limit the maximum number of attempts to 10. Table 1 includes a summary of the AUCCESS rate achieved by the different agents. Causal-PIK is 7 points higher than the best-performing baseline, requiring a lower number of attempts to solve the tasks.

**PHYRE:** We evaluate Causal-PIK on the PHYRE-1B Cross test set, which comprises 25 tasks distributed across 10 folds, with 10 variations per task and 10 tests per fold. For each test, we limit the maximum number of attempts to 100. As shown in Table 2, Causal-PIK achieves an AUCCESS rate over 10 points higher than the best-performing baseline. We also compare our results to baselines that use a reduced action space (Girdhar et al., 2020; Ahmed et al., 2021; Qi et al., 2021), which is guaranteed to lead to a 100 AUCCESS score for 10K actions if an optimal oracle is used.

As shown in Table 2, Causal-PIK performs comparably to baselines that utilize a drastically reduced action space. Our approach tackles a significantly harder problem by considering any point in the action space. Bakhtin et al. (2019) analyzed how the number of actions ranked by agents at test time affects performance. As shown in Figure 4 of their work, the AUCCESS of the DQN agent decreases as the number of ranked actions increases by orders of magnitude, up to a maximum of 100,000, which is still far from the 2,555,904 possible actions per puzzle in the full action space. We argue that discretizing the environment for action selection is an unrealistic constraint when aiming to develop generalist algorithms capable of solving complex physical reasoning tasks.

**Physics-Informed Kernel versus RBF Kernel:** We compare the posterior distributions of our Bayesian Optimization (BO) agent using the Physics-Informed Kernel and the RBF kernel. As shown in Figure 4, given the same set of initial observations, the BO Agent with the Physics-Informed Kernel produces a more expressive posterior that focuses on high-likelihood actions. Given two actions, one that moved the red ball further from the goal and another one that brought it closer but overshoots, the GP with the Physics-Informed Kernel constructs a posterior with three distinct regions: a

*Table 1.* AUCCESS scores of Causal-PIK, its RBF ablation variant, and state-of-the-art model on the Virtual Tools benchmark. Results are based on a maximum of 10 attempts per task, with higher scores indicating more efficient problem-solving.

| Model | AUCCESS ↑ |
|---|---|
| RAND | $16.0_{\pm 20.0}$ |
| DQN | $25.0_{\pm 24.0}$ |
| SSUP (Allen et al., 2020) | $58.0_{\pm 27.0}$ |
| Ours RBF | $42.0_{\pm 33.0}$ |
| Ours Causal-PIK | $\mathbf{65.0}_{\pm 25.0}$ |
| Humans (Allen et al., 2020) | $53.25_{\pm 23}$ |

*Table 2.* AUCCESS scores of Causal-PIK, its RBF ablation variant, and state-of-the-art models on PHYRE-1B Cross. Results are based on a maximum of 100 attempts per task and averaged over 10 test folds. Higher scores indicate more efficient problem-solving.

| Model | AUCCESS ↑ |
|---|---|
| Dec [Joint] (Girdhar et al., 2020) [*] | $40.3_{\pm 8}$ |
| MEM[†] | $18.5_{\pm 5.1}$ |
| DQN[†] | $36.8_{\pm 9.7}$ |
| Ahmed et al. 2021[†] | $41.9_{\pm 8.8}$ |
| RPIN (Qi et al., 2021)[†] | $42.2_{\pm 7.1}$ |
| RAND | $13.0_{\pm 5.0}$ |
| Harter et al. 2020 | $30.24_{\pm 8.9}$ |
| Ours RBF | $27.70_{\pm 9.68}$ |
| Ours Causal-PIK | $\mathbf{41.6}_{\pm 9.33}$ |
| Ours Causal-PIK @10[+] | $24.8_{\pm 9.22}$ |
| Humans @10[+] | $36.6_{\pm 10.2}$ |

[*] 1K reduced action space. [†] 10K reduced action space.
[+] max of 10 attempts per task.

red area on the top left side of the red ball, clustering actions that move the ball closer to the goal; a blue area on the top right, clustering actions that push it further away; and a yellow area representing unexplored actions that cause no movement. In contrast, the BO agent with the RBF kernel requires more steps to build an informative posterior.

**Human performance:** Participants found the puzzles from both benchmarks to be challenging. Causal-PIK achieved a higher AUCCESS score than humans on both benchmarks, except when it was restricted to 10 attempts in PHYRE. The high variance in AUCCESS rates for both humans and models suggests that the puzzles had mixed levels of difficulty. We computed AUCCESS scores across individual puzzles. On Virtual Tools, human per-puzzle scores were most correlated with SSUP ($r = 0.71$), followed by Causal-PIK ($r = 0.63$), then DQN ($r = 0.32$). Causal-PIK may be less correlated with humans but still have a higher overall

AUCCESS rate because it is able to solve several puzzles that humans find very difficult. This lowers the per-puzzle correlation, but highlights the overall performance of our method. On PHYRE, human scores were most correlated with Causal-PIK ($r = 0.73$), followed by Causal-PIK @10, which has limited attempts ($r = 0.71$), then the RBF kernel baseline ($r = 0.64$), and finally Harter et al. (2020) ($r = 0.55$). Overall, the high correlation in scores between humans and our model, even when restricted to a maximum of 10 attempts per puzzle, suggests high alignment in the types of physical dynamics that were found to be easy or difficult to reason about.

**Resistance to noisy predictions** We conduct an analysis to study the impact of the accuracy of the dynamics model on the performance of our method. To demonstrate that existing predictions are inherently noisy due to the characteristics of the train and test sets, we train the PHYRE dynamics model on tasks from the test templates, ensuring prior exposure to similar puzzles. As a result, the L2 error for object bounding boxes improved to 3.56, compared to 19.3 ± 4.55 when tested on entirely unseen puzzles.

Despite this difference in prediction accuracy, our method achieved an AUCCESS of 45, which is only 4 points higher than the 41.6 ± 9.33 AUCCESS we reported for the case with unseen dynamics. This demonstrates that even with a substantial increase in prediction error, the performance drop is small, indicating that our method remains resilient to noisy dynamic predictions. While improved dynamic predictions can enhance performance, our approach does not rely on perfect predictions, retaining robustness even in the presence of inaccuracies.

**On the shortcomings of RL agents:** Reinforcement learning (RL) agents like DQN have shown remarkable performance in strategic games such as Atari and Go after extensive training, but they struggle in tasks that require physical reasoning. In the context of the Virtual Tools and PHYRE physical reasoning tasks, DQN fails to generalize effectively to unseen puzzles due to its limited understanding of the physical properties of objects and the basic rules governing their interactions. Previous studies on the PHYRE and Virtual Tools benchmarks (Li et al., 2024; Allen et al., 2020) have underscored this limitation, demonstrating that RL agents struggle to understand the underlying physics involved beyond their strong mapping between states and actions. This shortcoming leads to a broad and inefficient unguided exploration process, requiring numerous attempts to discover promising areas. Furthermore, DQN fails to learn from past attempts, often repeating similar unsuccessful actions. In contrast, our approach focuses on learning from each failed attempt and uses the causality and physical insights from the Physics-Informed Kernel to guide exploration towards more promising areas.

## 6. Limitations

Causal-PIK currently does not share knowledge across tasks. Enabling agents to recognize similarities between tasks and leverage past observations from tasks requiring similar physical reasoning remains an avenue for future work. By identifying regions of the action space that share underlying dynamics, agents could integrate prior knowledge to solve new tasks more efficiently.

Another limitation arises from the noise introduced by causal effect predictions, which directly impacts performance. Poor predictions introduce misleading similarities, potentially guiding the agent in the wrong direction. Improving these predictions would improve the expressivity of the Physics-Informed Kernel. However, our results demonstrate that Causal-PIK remains robust despite this noise, suggesting potential for future sim-to-real transfer.

Additionally, the physical reasoning tasks considered in this study involve a three-dimensional action space. Scaling to higher-dimensional action spaces would require modifications to the causal effect predictor to accommodate the added complexity. However, the kernel equations (2-6) would remain unchanged, as the new dimensions would be encoded within the state returned by the model. Thereby, the fundamental structure of our approach remains unchanged, preserving the kernel's ability to compare the immediate effects of high-dimensional actions. Moreover, Bayesian Optimization (BO) is expected to remain effective in larger search spaces, as prior work has demonstrated its robustness in high-dimensional action spaces in real-world applications (Antonova et al., 2019).

## 7. Conclusion

We introduce Causal-PIK, a novel approach that integrates a Physics-Informed Kernel with BO to reason about causality in single-intervention physical reasoning tasks. By leveraging information from past failed attempts, our method enables agents to efficiently search for optimal actions, reducing the number of trials needed to solve tasks with complex dynamics. Our experimental results demonstrate that Causal-PIK outperforms state-of-the-art baselines, requiring fewer attempts on average to solve the puzzles from the Virtual Tools and PHYRE benchmarks.

## Acknowledgments

We thank Kelsey Allen and Kevin Smith for their insightful discussions and valuable feedback. We also thank the reviewers for their thoughtful feedback, which helped improve the quality of the manuscript. The Stanford Institute for Human-Centered Artificial Intelligence (HAI), the Sloan Research Foundation, and Intrinsic provided funds to support this work. TG was supported by a grant from Cooperative AI.

## Impact Statement

This paper presents work whose goal is to advance the field of Machine Learning. There are many potential societal consequences of our work, none which we feel must be specifically highlighted here.

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

## A. Dynamics Model

We train the dynamics models to predict causal effects of actions in the Virtual Tools and PHYRE benchmarks (Allen et al., 2020; Bakhtin et al., 2019). Our model is based on the Region Proposal Interaction Networks (RPIN) architecture (Qi et al., 2021) and is trained on puzzles that share similar underlying physical principles with the test puzzles.

For the Virtual Tools benchmark, we generate 10 variations of each of the 20 original puzzles, modifying object sizes and relative positions to create diverse scenarios (see Figure 5). None of the original puzzles are included in training. For each puzzle variation, 300 actions are generated, distributed evenly across the three action object types. At least 50% of actions result in collisions between the action object and a dynamic object, while 10% simulate the absence of an action object in the environment. The remaining actions are randomly sampled using a Sobol generator to determine object placement. The inclusion of "no-action" cases enables the model to implicitly learn that stationary objects remain static when no net external force is applied.

For the PHYRE benchmark, we train 10 separate dynamics models, one per fold. Each model is trained on 20 out of the 25 puzzles assigned to the training set for that fold. For each puzzle, we generate 500 actions, of which 350 result in failed rollouts, 150 result in successful rollouts, and 50 simulate the absence of an action object. Actions are randomly drawn from the 100,000 pre-selected actions provided by (Bakhtin et al., 2019).

At inference time, the model receives an image of the puzzle, along with the initial bounding boxes of all dynamic objects and the action object. The model outputs the bounding boxes for the next $n_{\text{pred}}$ time steps. We set $n_{\text{pred}}$ to 20, which usually captures one collision but not the full roll-out. The choice of $n_{\text{pred}}$ helps the dynamic model focus on learning the immediate causal effect of an action, which is then used to compare similarity between actions.

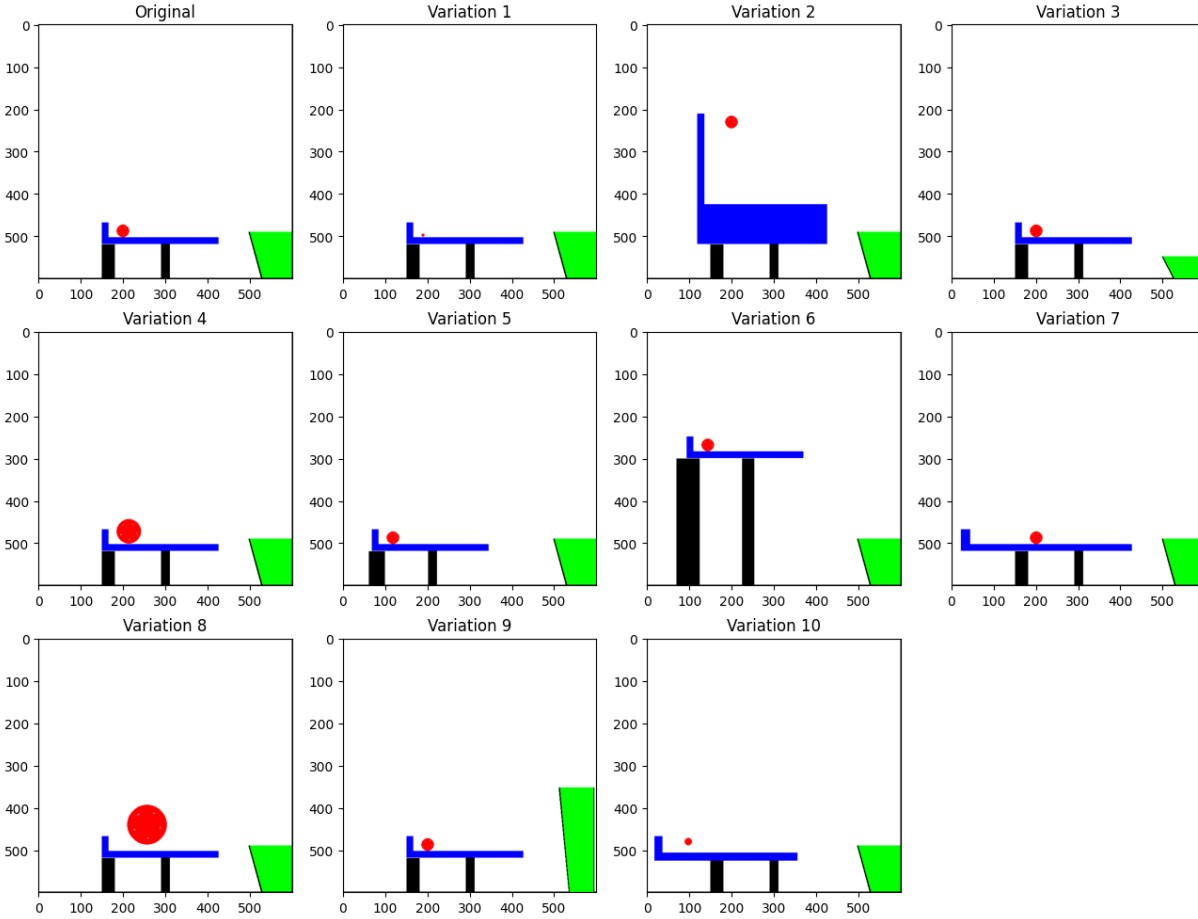

*Figure 5.* Puzzle variations for one of the original puzzles in the Virtual Tools benchmark. The puzzle variations share a similar underlying physical concept with the original puzzle, but have different dynamics due to varying object sizes and relative positions.

# B. PHYRE Human Experiment

We adapted the PHYRE-1B benchmark into a suite of online games using Planck.js (Shakiba, 2017), a JavaScript rewrite of the Box2D physics engine used in PHYRE (Bakhtin et al., 2019). The experiment was posted on Prolific, an online crowd-sourcing research platform. We recruited $n = 50$ participants (*age*: M = 37, SD = 11; *gender*: 20 female, 27 male, 1 non-binary, and 2 undisclosed; *race*: 29 White, 9 Black, 5 Asian, 1 American Indian/Alaska Native, 4 Multiracial, and 2 undisclosed) and compensated them at a rate of $12/hour. All participants were native English speakers residing in the US.

Participants were first given instructions for the task, including a "playground" environment where they learned about the different objects and the dynamics of the world. They then solved a simple practice puzzle before moving on to the main puzzles. During the main portion of the experiment, they were given one variation of each of the 25 puzzles in Phyre-1B in a random order. On each puzzle, participants were initially presented with a freeze-frame of the scene. They attempted to solve the puzzle by using the mouse to draw and place the action object (a red ball) in any valid location in the scene. On each attempt, they watched the simulation run until it either succeeded, after which they continued on to the next puzzle, or timed out or all objects stopped moving, at which point they could try again. If they ran out of attempts (maximum 10), then they were also directed to the next puzzle. We enforced all the same physics parameters and time limit described in the original PHYRE benchmark. We recorded participants' actions ($x_{pos}$, $y_{pos}$, and $action_{obj} = r_{ball}$) on each attempt. Overall, participants spent an average of 1.7 minutes (SD = 1.4) on each puzzle.

