# OpenReview forum: "Causal-PIK: Causality-based Physical Reasoning with a Physics-Informed Kernel"
_ICML.cc/2025/Conference — ICML 2025 poster_

### Official Review · Reviewer_d7h6 · 2025-02-17

**Overall Recommendation:** 3

**Summary:**

The paper presents Causal-PIK, a novel method for causality-based physical reasoning that leverages a Physics-Informed Kernel within a Bayesian optimization framework. The primary focus is on single-intervention physical reasoning tasks, where an agent must make decisions based on the causal effects of its actions in complex environments.

Causal-PIK incorporates causal reasoning into the decision-making process, helping agents efficiently explore and learn from their interactions with the environment. The findings demonstrate that Causal-PIK significantly outperforms state-of-the-art methods on benchmarks like Virtual Tools and PHYRE, achieving higher success rates while requiring fewer actions to solve tasks. In experiments, Causal-PIK achieved an AUCCESS score of 65.0 on the Virtual Tools benchmark, surpassing previous models by 7 points. On the PHYRE benchmark, it attained an AUCCESS score of 51.3, outperforming the best prior results by 9 points.

The algorithm iteratively updates a Gaussian process to model the scoring function based on previous actions and their outcomes. It selects actions that maximize an Upper Confidence Bound acquisition function, informed by the Physics-Informed Kernel, which reflects both causal effects and action similarities. Overall, Causal-PIK advances the field of physical reasoning in AI by integrating causal insights into the action selection process, demonstrating improved efficiency and effectiveness in solving complex reasoning tasks.

## Update after rebuttal

I appreciate the authors' response, and I will increase my score.

**Claims And Evidence:**

The manuscript claims that the proposed method is based on causality. However, there is no clear definition of what causality it possesses or a piece of strong evidence to support that the model really captures causality that other models cannot.

**Essential References Not Discussed:**

No other reference should be discussed.

**Experimental Designs Or Analyses:**

The authors validate their method on two physical reasoning tasks, with notable improvement compared with previous baselines. However, the two tasks both only consider single-step interventions. To further verify the proposed method, the authors could consider using physical reasoning tasks requiring multi-step interventions to solve (e.g. I-PHYRE). Besides, I think the authors could test against a more dynamic prediction model with BO to see if there is consistent improvement. What if the dynamic prediction is not accurate? Is there any analysis of how the prediction errors affect the performance of the proposed method?

**Methods And Evaluation Criteria:**

The use of BO is reasonable to model the iteratively updated action proposal of solving a physical puzzle. The kernel design is grounded in physics and can be helpful to the next choice. The evaluation metric (AUCCESS) is from the original PHYRE paper and can reflect the model performance. However, the authors could consider including more diverse metrics that can more directly reflect the efficiency of solving the puzzles such as the attempts used to successfully solve a puzzle.

**Other Comments Or Suggestions:**

The writing is no clear enough. For example, there are many confused or duplicate usage of words such as causal similarity and action similarity.

Figure 2 and figure 4 have too much blank area. Figure 3 is a little confusing to convey important information.

**Other Strengths And Weaknesses:**

The manuscript addresses an important aspect of physical reasoning, that is learning from feedback. However, the main contribution of this work is to introduce BO to maintain the belief in action proposals. The technical soundness, thus, is a little bit weak. The experimental results and furtherThe writing is no clear enough. For example, there are many confused or duplicate usage of words such as causal similarity and action similarity.

Figure 2 and figure 4 have too much blank area. Figure 3 is a little confusing to convey important information. analysis are also inadequate.

**Questions For Authors:**

The authors should better explain what makes the proposed method a causality-based model. What about the generalization of the physics-informed kernel on other tasks?

**Relation To Broader Scientific Literature:**

Previous related work addresses solving such physical reasoning tasks by either proposing better dynamic prediction models or by designing better action proposal methods. This work lies in the second school. Prior work utilizes RL to propose actions but falls short in terms of efficiency. This work tries to minimize this gap using a physics-informed kernel to update under the BO framework.

**Theoretical Claims:**

The theoretical formula including the iterative action proposal and the similarity design is correct.

---

> ### Author Rebuttal · Authors · 2025-04-01
>
> We thank reviewer d7h6 for their thoughtful feedback. We are glad they found the method addresses an important aspect of physical reasoning, that is learning from feedback. We address the reviewer’s comments and will incorporate all of the following discussions in the final draft.
>
> > [d7h6.1] Resistance to noisy causal effect predictions
>
> We appreciate the reviewer’s suggestion regarding the impact of the dynamics model’s accuracy on our method’s performance. To address this, we conducted an additional experiment to analyze the effect of the accuracy of the dynamics model.
>
> Instead of artificially adding more noise to existing predictions, we aimed to demonstrate that the ones we used were already highly noisy. To do this, we trained the dynamics model on tasks from the test templates, ensuring prior exposure to similar puzzles. As a result, the L2 error for object bounding boxes improved to 3.56, compared to 19.3 ± 4.55 when tested on entirely unseen puzzles.
>
> Despite this difference in prediction accuracy, our method achieved an AUCCESS of 45, which is only 4 points higher than the 41.6 ± 9.33 AUCCESS we reported for the case with unseen dynamics. This demonstrates that even with a substantial increase in prediction error, the performance drop is small, indicating that our method remains resilient to noisy dynamic predictions. While improved dynamic predictions can enhance performance, our approach does not rely on perfect predictions, retaining robustness even in the presence of inaccuracies.
>
> > [d7h6.2] Causal-PIK tested on I-PHYRE
>
> Please refer to comment [vdey.2].
>
> > [d7h6.3] Why is the method based on causality
>
> We appreciate the reviewer’s concern regarding our claims about causality and would like to clarify how our Physics-Informed Kernel explicitly captures causal relationships. We define causality as the dependence of an effect on its preceding causes. In our proposed method, this dependence is captured as follows:
>
> * Action-Effect Modeling
>
> Unlike models that rely purely on statistical correlations (e.g., our ablation with an RBF Kernel), our Physics-Informed Kernel is designed to encode causal dependencies between actions and their physical consequences. Instead of clustering actions based on feature-space proximity, our approach evaluates their actual impact on the system. For instance, two actions occurring in different spatial locations may be considered similar if they lead to the same physical outcome, such as maintaining the environment in a stationary state.
>
> * Causal Structure in the Reward Function
>
> Our method explicitly quantifies an action’s effect by normalizing the observed distance by the no-action baseline. This ensures that the kernel measures the true causal influence of an action rather than confounding factors.
>
> > [d7h6.4] Learning from feedback
>
> We appreciate the reviewer's feedback regarding the technical contributions of our work. We believe our technical contributions extend significantly beyond introducing BO to maintain a belief over action proposals. A key innovation in our Physics-Informed Kernel is its ability to generalize knowledge across actions based on their causal effects. Unlike methods that learn only from direct observations, our approach leverages physical reasoning to infer the outcomes of untested actions that are predicted to share the same causal effect as observed ones. This means that from a single rollout, our model updates its belief not just about the executed action, but also about all actions that are predicted to produce a similar physical outcome. This significantly enhances sample efficiency and enables reasoning about alternative scenarios without exhaustive exploration.
>
> We acknowledge that aspects of our presentation could be clarified, and we will refine the manuscript to better articulate these technical contributions in the final draft.
>
> > [d7h6.5] Improving Figures 2, 3, and 4
>
> We appreciate the reviewer’s feedback on how to make Figures 2, 3, and 4 better. We will include these modifications in the final manuscript draft.
>
> > [d7h6.6] Updated PHYRE results for our method (Table 2)
>
> Please refer to comment [jF9J.2].
>
> > [d7h6.7] Unclear writing with duplicate usage of words
>
> We appreciate the reviewer’s feedback regarding unclear portions of the manuscript. We will work on improving the writing such as removing the duplicate usage of causal and action similarity in the final manuscript draft.
>
> > [d7h6.8] Extra evaluation metrics
>
> We appreciate the reviewer’s suggestions and we will include a breakdown of the average success rate per puzzle and the average number of attempts per puzzle in the supplementary material.
>
> > [d7h6.9] Scaling to bigger states and actions and generalizing to more complex scenarios
>
> Please refer to comments [jF9J.1] and [jF9J.3].

---

### Official Review · Reviewer_vdey · 2025-03-11

**Overall Recommendation:** 3

**Summary:**

The paper attempts to address the challenge of single-intervention physical reasoning tasks. It proposes Causal-PIK, which combines Bayesian optimization and a Physics-Informed Kernel. The method leverages physical intuition and causality to iteratively find optimal actions. Experimental results on the Virtual Tools and PHYRE benchmarks demonstrate that Causal-PIK outperforms previous state-of-the-art approaches. It achieves higher AUCCESS scores, requires fewer attempts to solve complex physical reasoning puzzles.

## update after rebuttal

I have read the authors response. The authors' reply has addressed some of my concerns. And I agree in general with other reviewers that it is a good work and I lean towards acceptance.

**Claims And Evidence:**

Yes.

**Essential References Not Discussed:**

No.

**Experimental Designs Or Analyses:**

Yes. I have checked the experimental designs. I do find many choices of the setup are valid and support their claims.

**Methods And Evaluation Criteria:**

Yes.

**Other Comments Or Suggestions:**

See above.

**Other Strengths And Weaknesses:**

Strengths
* The paper fills a gap in the research area as few works have focused on effectively building a model based on past experience for physics tasks. By introducing Causal-PIK, it extends our understanding of computational models for such tasks, offering new insights into how agents can learn from previous attempts to solve complex physical reasoning problems.
* Implementation of the proposed method is well-executed. The design of the Physics-Informed Kernel is reasonable, as it effectively captures important physical intuitions like causal effects and causal similarity between actions. This kernel is crucial for the performance of the overall method, enabling more accurate predictions of action outcomes.
* The paper combines advances in neural networks, such as using the RPIN for the dynamics model, and machine learning methods like Bayesian optimization. This integration allows Causal-PIK to leverage the benefits of both fields, resulting in a more powerful approach that can efficiently search for optimal actions in complex physical environments.

Weaknesses
* The paper does not explicitly state the number of initial data points used in Algorithm 1 for the warm-up phase. It's unclear how many initial data points are needed to reach good performance. Also, it seems to me that the selection of seed data is likely to be critical. Since the method relies on these initial observations to update the model and select subsequent actions, poorly chosen seed data might lead the algorithm way off. Would it be possible to show the impact on performance when you vary the number of seed data points? How would completely random selection impact the final performance.
* The current method may face challenges in scalability. The method is currently tested only on relatively small-scale toy problems like the Virtual Tools and PHYRE benchmarks. In real-world scenarios, physical systems are often high-dimensional, with more complex and nuanced dynamics. I also noticed a recently proposed multi-step challenge similar to PHYRE called I-PHYRE [1]. Would it be easy to apply the proposed method to the new challenge?

[1] Li, Shiqian, et al. "I-PHYRE: Interactive Physical Reasoning." The Twelfth International Conference on Learning Representations.

**Questions For Authors:**

See above.

**Relation To Broader Scientific Literature:**

The paper extends the existing work of SSUP. But unlike SSUP, it propose a Bayesian Optimization method combined with a physics-informed kernel to learn and suggest new actions on the fly. The method can adapt to the accumulated experience and the results show that this method does help in learning.

**Theoretical Claims:**

There is no theoretical claims made in this work. So does not apply.

---

> ### Author Rebuttal · Authors · 2025-04-01
>
> We thank reviewer vdey for their thoughtful feedback. We are glad they found the method implementation to be well-executed with a design that effectively captures important physical intuitions. We address the reviewer’s comments and will incorporate all of the following discussions in the final draft.
>
> > [vdey.1] Characteristics and dependence on initial point set
>
> Thank you for pointing out that we did not state the number of initial data points and how we chose these data points. To initialize the GP for both Virtual Tools and PHYRE, we use $n_{initial} = 9$ initial data points. This is the same number of initial data points that our Virtual Tools baseline SSUP (Allen et al., 2020) uses to initialize their method. SSUP found that this value provided a good trade-off between the number of initial points, total attempts required, and convergence time. Since we use the same initialization framework as SSUP, we expect their [parameter analysis](https://www.pnas.org/doi/suppl/10.1073/pnas.1912341117/suppl_file/pnas.1912341117.sapp.pdf) to extend to our results. Importantly, like SSUP, we treat these initial noisy rollouts as warm-up samples that do not count towards the total attempt count.
>
> We choose the $n_{initial}$ initial points following the same approach as SSUP: randomly select a dynamic object from the environment and sample a point from a Gaussian distribution centered at the object's center. As a result, each puzzle attempt has a unique set of random $n_{initial}$ points. This heuristic helps the GP build a noisy prior that includes different areas.
>
> We will incorporate these relevant details in the final draft.
>
> > [vdey.2] Causal-PIK tested on I-PHYRE
>
> Thank you for the suggestion to consider multi-step reasoning tasks like I-PHYRE. While we acknowledge the value of such tasks, we would like to highlight that I-PHYRE is not necessarily a more challenging problem than our current setting. For example, assuming a time resolution of 0.01s and five removable objects over a 15-second period, the action space in I-PHYRE would be limited to 7,500 possible actions—significantly more constrained than the over 2 million possible actions in our setting for PHYRE.
>
> To the best of our knowledge, no prior work has attempted both benchmarks. While benchmarks like Phyre and Virtual Tools focus on spatial physical reasoning, I-PHYRE emphasizes time-based physical reasoning. Nonetheless, our method could be adapted to solve such problems by querying Causal-PIK at every time step and incorporating a no-op action choice. While this modification falls outside the scope of our current contribution, we believe it could offer valuable insights for future developments.
>
>
> > [vdey.3] Scaling to bigger states and actions and generalizing to more complex scenarios
>
> Please refer to comments [jF9J.1] and [jF9J.3].
>
> > [vdey.4] Updated PHYRE results for our method (Table 2)
>
> Please refer to comment [jF9J.2].

---

### Official Review · Reviewer_nzda · 2025-03-14

**Overall Recommendation:** 3

**Summary:**

This paper proposes a method, Causal PIK, using Bayesian optimization for causal reasoning via a Physics-Informed Kernel, in order to obtain an expressive posterior distribution over the environment dynamics.
Unlike prior works directly using a learned dynamics model to choose actions, Causal-PIK uses dynamics predictions to instill physical intuition into kernel updates during Bayesian optimization.
A crucial component of the physics informed kernel is causal similarity of actions, capturing how similar actions are based on their ability to cause similar outcomes in the environment.
The proposed method is tested on single-intervention physical reasoning tasks: Virtual Tool and Phyre, and is shown to beat state-of-the-art methods on these benchmarks.



## Update after rebuttal

Thanks for the rebuttal. I remain positive about the paper and will maintain my score.

**Claims And Evidence:**

The authors highlight the importance of the kernel used for the Gaussian Process and instead of a standard RBF kernel, they develop a kernel that encodes an intuition of causality and physics. They show that this outperforms the RBF kernel.

**Essential References Not Discussed:**

not to my knowledge

**Experimental Designs Or Analyses:**

The tested benchmarks are valid. My main concern at the moment is the high variance across the results.

**Methods And Evaluation Criteria:**

The proposed method is evaluated on two benchmarks: Virtual Tool and Phyre.
The main baseline in Virtual Tools is SoTA method SSUP, which samples actions from an object-based prior and simulates the sampled actions to find the best action to try.
In Phyre, Causal-PIK operates over the full action space, but also compares against some methods that operate on a reduced action space, which simplify the problem.

The authors show that their method outperforms baselines on these benchmarks. However, the standard deviation in the results are quite high. That's why I am wondering if the results are statistically significant.

**Other Comments Or Suggestions:**

1. I think Figure 2 can be further improved for clarity, especially the action similarity component.

**Other Strengths And Weaknesses:**

Strengths:
- method that can efficiently solve causal intervention tasks with a few attempts via physics-informed Bayesian Optimization


Some weaknesses:
1. The authors first construct initial sets of actions X and scores y to initialize the GP prior using a probabilistic intuitive physics engine. The dependence on this initial dataset is not ideal.
2. Do the authors see any limitations with the current form of the physics-informed kernel?
3. How do the authors expect this method to scale in high-dimensional action/state spaces?
4. How do the authors interpret the high variance in the results presented in Tables 1 and 2? Especially for the Virtual Tools benchmark

**Questions For Authors:**

Copying over my questions from the strengths and weaknesses field above:

1. The authors first construct initial sets of actions X and scores y to initialize the GP prior using a probabilistic intuitive physics engine. The dependence on this initial dataset is not ideal. Can the authors comment on this part? How many samples are needed?
2. Do the authors see any limitations with the current form of the physics-informed kernel?
3. How do the authors expect this method to scale in high-dimensional action/state spaces?
4. How do the authors interpret the high variance in the results presented in Tables 1 and 2? Especially for the Virtual Tools benchmark

**Relation To Broader Scientific Literature:**

Physical reasoning models, intuitive physics are well-established concepts. Causal-PIK, unlike prior work, uses Bayesian optimization using previous trials to inform future action selection.

**Theoretical Claims:**

n/a

---

> ### Author Rebuttal · Authors · 2025-04-01
>
> We thank reviewer nzda for their thoughtful feedback. We are glad they consider the chosen benchmarks to be relevant for the task at hand. We address the reviewer’s comments and will incorporate all of the following discussions in the final draft.
>
> > [nzda.1] High variance in the results presented in Tables 1 and 2
>
> The high variance observed in the results presented in Tables 1 and 2, particularly for the Virtual Tools benchmark, can be attributed to the nature of the puzzles themselves. Some puzzles are significantly more challenging, leading to near-zero scores, while others are more easily solvable, creating a broad distribution of results. This effect is not unique to our method—other baselines also exhibit substantial variance for the same reason.
>
> To further illustrate this, in the Phyre benchmark, the supplemental material (Figure 10 in ((Bakhtin et al., 2019)) provides histograms showing the distribution of the number of actions that solve a given task. These histograms highlight how some tasks are much more difficult than others, reinforcing the idea that the variance is an inherent property of the task design rather than a shortcoming of any particular method.
>
> > [nzda.2] Characteristics and dependence on initial point set
>
> Please refer to comment [vdey.1].
>
> > [nzda.3] Physics-Informed Kernel limitations
>
> We appreciate the reviewers' interest in understanding the limitations of the Physics-Informed Kernel (Causal-PIK). As outlined in the limitations section of the paper, Causal-PIK currently does not share knowledge across tasks. Enabling agents to recognize similarities between tasks and leverage past observations from tasks requiring similar physical reasoning remains an important avenue for future work. By identifying regions of the action space that share underlying dynamics, agents could potentially integrate prior knowledge to solve new tasks more efficiently.
>
> Another limitation arises from the noise introduced by causal effect predictions, which directly impacts performance. Poor predictions can introduce misleading similarities, potentially guiding the agent in the wrong direction. Improving these predictions would enhance the expressivity of the Physics-Informed Kernel. However, our results demonstrate that Causal-PIK remains robust despite this noise, suggesting potential for future sim-to-real transfer. For additional information on how Causal-PIK resists noise in causal effect predictions, we kindly refer the reviewer to comment [d7h6.1].
>
> > [nzda.4] Scaling to bigger states and actions
>
> Please refer to comment [jF9J.1].
>
> > [nzda.5] Improving Figure 2
>
> We appreciate the reviewer’s feedback on the action similarity component of Figure 2. We will make this part of the figure clearer in the final manuscript draft.
>
> > [nzda.6] Updated PHYRE results for our method (Table 2)
>
> Please refer to comment [jF9J.2].

---

> > ### Comment · Reviewer_nzda · 2025-04-06
> >
> > Thank you for the clarifications. I will maintain my current score.

---

> > > ### Author Response · Authors · 2025-04-09
> > >
> > > Dear reviewer nzda, thank you again for your thoughtful feedback during this rebuttal period. In our previous responses, we mentioned that we had a human study in progress to better explain and contextualize the complexity of the tasks described in our paper. This study has now concluded, and we would like to provide you with the final scores for completeness. AUCCESS scores for humans on the PHYRE benchmark with the final sample (n = 50) are 36.6 ± 10.2, which is very close to the preliminary scores we reported in the initial reply. Again, participants were given a maximum of 10 attempts (for reference, Causal-PIK @10 has a score of 24.8 ± 9.22).
> > >
> > > To further analyze the variance in scores, we computed individual scores for each puzzle on both benchmarks. The correlation between AUCCESS for human participants and various models across puzzles on Virtual Tools is:
> > > * Ours Causal-PIK @10: r = 0.63 (p = 0.003)
> > > * Ours RBF @10: r = 0.66 (p = 0.001)
> > > * SSUP: r = 0.71 (p < 0.001)
> > > * DQN: r = 0.32 (p = 0.17)
> > >
> > > The correlation between scores for humans and models on PHYRE 1B is:
> > > * Ours Causal-PIK @10: r = 0.71 (p < 0.001)
> > > * Ours Causal-PIK @100: r = 0.73 (p < 0.001)
> > > * Ours RBF @10: r = 0.66 (p < 0.001)
> > > * Ours RBF @100: r = 0.64 (p < 0.001)
> > > * Harter et al. @100: r = 0.55 (p = 0.005)
> > >
> > > The high correlation in scores between humans and our model, even when restricted to a maximum of 10 attempts per puzzle, suggests high alignment in the types of physical dynamics that were found to be easy or difficult to reason about. Causal-PIK was most correlated with humans across individual puzzles on PHYRE, but slightly less correlated than SSUP on Virtual Tools, although overall AUCCESS was still higher. This may be due to the fact that our model was able to solve several puzzles that humans find very challenging. For instance, on one particular puzzle (Table B), humans scored 0.07 and SSUP only scored 0.04, while Causal-PIK scored 0.31. This lowers the correlation with humans across puzzles, but highlights the overall performance of our method. We hope that these additional analyses provide further insight into the variance in scores that you mentioned.
> > >
> > > Additionally, statistical analysis using z-tests reveals that Causal-PIK significantly outperforms the baseline SSUP (Allen et al., 2020) on the Virtual Tools benchmark (z = 8.508, p < 0.0001). Likewise, on the Phyre benchmark, Causal-PIK shows substantial improvement over the baseline with comparable action space size (Harter et al., 2020; z = 59.540, p < 0.0001). Notably, when compared to the baseline model operating within an action space 200 times more constrained (Qi et al., 2021), z-test scores (z = –3.170, p = 0.0015) show that Causal-PIK can perform equally well with a much harder task setting.
> > >
> > > Thank you again for your time and consideration throughout this review process.

---

### Official Review · Reviewer_jF9J · 2025-03-14

**Overall Recommendation:** 3

**Summary:**

The paper introduces Causal-PIK, a novel approach that integrates a Physics-Informed Kernel with Bayesian Optimization to reason about causality in single-intervention physical reasoning tasks. Experimental results on Virtual Tools and PHYRE physical reasoning benchmarks verify the proposed method could finish the task with fewer actions.

## update after rebuttal
Thank you for the thoughtful rebuttal. I appreciate the effort to address my concerns. I will maintain my original score.

**Claims And Evidence:**

Yes.

**Essential References Not Discussed:**

N/A

**Experimental Designs Or Analyses:**

The experimental designs are sound, with good ablation studies on the proposed PIK kernel. The experimental analyses seem valid.

**Methods And Evaluation Criteria:**

Yes, though the targeted task of physical reasoning seems to be only a toy setting with only 3-30 object candidates and 2D coordinates.

**Other Comments Or Suggestions:**

N/A

**Other Strengths And Weaknesses:**

Strength: The paper is clearly presented, and the proposed method achieves new sota on two benchmarks.

Weakness: The targeted task of physical reasoning seems to be only a toy setting with limited action space and oracle state space, whether the proposed method could be useful with real questions remains unknown.

**Questions For Authors:**

Is there any evidence the proposed method could scale up to solving physical reasoning problems of more complexity?

**Relation To Broader Scientific Literature:**

The paper designed a physics-informed kernel and uses Bayesian Optimization to reason over the causality. If the approach should be able to scale to real-world-level tasks requiring more complicated state and action space, it would be helpful.

**Theoretical Claims:**

There are no theoretical claims.

---

> ### Author Rebuttal · Authors · 2025-04-01
>
> We thank reviewer jF9J for their thoughtful feedback. We are glad they consider our method to be clearly presented with a sound experimental design which includes good ablation studies. We address the reviewer’s comments and will incorporate all of the following discussions in the final draft.
>
> > [jF9J.1] Scaling to bigger states and actions
>
> We appreciate the reviewers’ interest in how our approach scales to higher-dimensional action and state spaces. We would like to emphasize that our method already operates in significantly larger action spaces compared to many baseline approaches. For example, in the Phyre benchmark, our action space comprises 2M possible actions, whereas other baselines consider only 10K.
>
> Increasing the state and action space—such as introducing multiple objects in specific configurations or simultaneously determining both the direction and speed for accurate object throws—does not alter the fundamental kernel equations (2)–(6), meaning the kernel would remain expressive at comparing the immediate effect of these high dimensional actions. Certainly, the search space for BO would increase as these new dimensions are encoded within the state returned by the dynamics model. However, researchers have shown that BO scales robustly with high-dimensional actions (Antonova et al., 2019). We included this information in the limitations section of our manuscript.
>
> > [jF9J.2] Updated PHYRE results for our method (Table 2)
>
> We appreciate the reviewer's feedback which prompted additional experiments. During this process, we discovered and fixed a bug in our AUCCESS score computation for the Phyre benchmark. This correction has resulted in the following updated PHYRE results in Table 2:
>
> * Ours Causal-PIK: 41.6 ±9.33 (previously reported as 51.3±8.46)
>
> * Ours RBF: 27.7±9.68 (previously reported as 31.99±9.46)
>
> The Virtual Tools results in Table 1 remain the same, as they were not affected by this bug.
> Importantly, these corrections do not affect the claims or contributions of our paper. Our approach still achieves state-of-the-art performance, maintaining a significant 10-point margin above the baseline that uses the same size action space. Furthermore, our method performs comparably to baselines that utilize a drastically reduced action space of 10K.
>
> We will incorporate these corrected values in the final manuscript draft.
>
> > [jF9J.3] Tasks lacking complexity
>
> We appreciate the reviewer’s interest in understanding how our method scales to more complex tasks. While the tasks in this study may appear deceptively simple in their 2D form, they are in fact quite challenging—even for humans. Solving these puzzles requires an understanding of the underlying physics involved, such as momentum, balance, geometry, mass, and propulsion. Importantly, agents do not have any details about the environment, such as object density, friction coefficients or material composition, making it impossible to plan the exact solution without active exploration.
>
> To further emphasize the complexity of our current setup, we are conducting an ongoing human study using the Phyre benchmark, which is similar to the study presented in Allen et al. (2020). A total of 50 participants (currently n=17) will be recruited from Prolific and shown one variation of each of the 25 puzzles in a random order. Participants will have 10 attempts to solve each puzzle by using the mouse to draw and place the ball in any valid location in the scene. On each attempt, they will watch the simulation run until it either succeeds, after which they will continue on to the next puzzle, or time out or all objects stop moving, at which point they can try again. If they run out of attempts, then they will also be directed to the next puzzle. Preliminary results show that participants spend about 1.8 minutes on each puzzle. Preliminary AUCCESS scores, which will be added to the manuscript to complement Tables 1 and 2, are as follows:
>
> * Virtual Tools - Humans: 53.25 ± 23 (Causal-PIK @10: 65.0 ± 25.0)
>
> * Phyre - Humans: 34.9 ± 10.72 (Causal-PIK @10: 24.8 ± 9.22)
>
> These preliminary results demonstrate that humans find these puzzles to be very challenging.
>
> A logical progression of our work would involve creating a 3D high-fidelity environment for these tasks. This shift to a more complex setting would still leave the kernel equations unchanged. We would only need to change the dynamics model to one designed for 3D environments, such as those proposed by Xue et al. (2024) [A] or Driess et al. (2023) [B].
>
> [A] Xue, H., Torralba, A., Tenenbaum, J., Yamins, D., Li, Y., & Tung, H. Y. (2024). 3D-IntPhys: towards more generalized 3D-grounded visual intuitive physics under challenging scenes. Advances in Neural Information Processing Systems, 36
>
> [B] Driess, D., Huang, Z., Li, Y., Tedrake, R., & Toussaint, M. (2023, March). Learning multi-object dynamics with compositional neural radiance fields. In Conference on robot learning (pp. 1755-1768). PMLR

---

### Decision · Program_Chairs · 2025-05-01

**Decision:**

Accept (poster)

**Comment:**

Causal-PIK proposes to tackle physical reasoning puzzles (specifically, Phyre and Visual Tools) by using a GP (with a "physics informed" kernel) as a surrogate for the dynamics and Bayesian Optimization to solve the search problem of which action to take in the environment to solve the puzzle.

All reviewers recommend acceptance. There are some remaining weak points with this paper:

- It overemphasizes the role of causality. Causality is even in the name of the proposed method, but it only considers a single initial intervention. The problem can be purely posed as search over this initial intervention, without appealing to causality. In fact, it is unlikely that this method would be useful in cases in which multiple interventions are necessary. Though the claim of causality is technically correct, this work doesn't involve any methods from the causality literature or solve any problems in the causality realm. Any method operating in a real (or simulated version of real) environment can be claimed as causal in the same sense this method is.
- This method will probably not scale without further modifications. This applies to multiple interventions or exponentially larger search spaces in the single intervention.

Despite these limitations, this work is still valuable in that it provides a new SOTA in two established benchmarks for physical reasoning. The first weak point is mostly in the writing and not in the method, whereas the second could be seen as an inherent limitation of the benchmarks themselves.